# Psychological and Mental Sequelae in Elite Athletes with Previous SARS-CoV-2 Infection: A Systematic Review

**DOI:** 10.3390/ijerph192416377

**Published:** 2022-12-07

**Authors:** Valerio Flavio Corona, Maria Rosaria Gualano, Maria Francesca Rossi, Angelica Valz Gris, Carlotta Amantea, Umberto Moscato, Walter Ricciardi

**Affiliations:** 1Department of Life Sciences and Public Health, Section of Hygiene, Università Cattolica del Sacro Cuore, Largo Francesco Vito 1, 00168 Rome, Italy; 2School of Medicine, UniCamillus-Saint Camillus International University of Health Sciences, 00131 Rome, Italy; 3Leadership in Medicine Research Center, Università Cattolica del Sacro Cuore, 00168 Rome, Italy; 4Center for Global Health Research and Studies, Università Cattolica del Sacro Cuore, Largo Francesco Vito 1, 00168 Rome, Italy; 5Department of Life Sciences and Public Health, Section of Occupational Health, Università Cattolica del Sacro Cuore, Largo Francesco Vito 1, 00168 Rome, Italy

**Keywords:** COVID-19, post-acute sequelae of SARS-CoV-2 infection, mental health, psychological stress, elite athletes

## Abstract

During the COVID-19 pandemic, many athletes from several sporting disciplines were infected with the SARS-CoV-2. The aim of this systematic review is to summarize the current scientific evidence on the psychological sequelae and mental health of elite athletes who have been infected by the virus. The review was performed following the Preferred Reporting Items for Systematic Reviews and Meta-Analyses (PRISMA) Statement; three databases were searched: PubMed, ISI Web of Knowledge, and Scopus. The initial search resulted in 2420 studies; after duplicate removal and screening by title and abstract, 41 articles were screened by full-text. A total of four eligible articles were included in the review. All included articles measured depression and anxiety in athletes who had suffered from COVID-19, while in three papers levels of stress were measured. Overall, the only two questionnaires used in more than one study were the DASS-21 and the APSQ. In our systematic review, we highlighted that mental and psychological health in elite athletes has the same importance as physical health. This statement suggests that these examinations should be introduced and performed during the competitive sports’ medical examinations conducted at the start of the sporting season, which currently consists only of the examination of physical parameters. Due to lack of studies on the topic, the results of our review show that mental health in athletes with a history of SARS-CoV-2 infection is an issue that requires more investigation, considering the evidence of clinical consequences. The importance of post-infection psychological sequelae is significant in assessing possible repercussions on the athletes’ sporting performance.

## 1. Introduction

Long COVID-19 is a condition that occurs in individuals with a probable or confirmed history of SARS-CoV-2 infection, starting 3 months after primary infection, which persists for at least 2 months and cannot be explained by an alternative diagnosis [1]. A variety of symptoms can be observed, exacerbating the signs of primary infection or characterised by the onset of new conditions, but generally patients report a decrease in physical and psychological performance [2].

Elite athletes, representing the category of the highest level of competitiveness [3], have an increased risk of virus infection, with the consequences that follow, due to increased exposure to pathogens (worldwide travel/international competitions) or an impaired immune system (training continues during infection/training starts immediately after infection, stressful training or competition and exercise in extreme climatic conditions) [4].

During the COVID-19 pandemic, many athletes from various sporting disciplines were infected with the SARS-CoV-2 virus. Among the complications due to the infection, prominent are the cardiovascular ones, in particular myocarditis and pericarditis [5]. Considering the high incidence of cardiological diseases in elite athletes [6], specific attention has been paid to the screening of professional athletes who have been infected with SARS-CoV-2, before their return to sporting activity [7,8]. The Italian Sports Medical Federation (FMSI) established a protocol for testing athletes in order for them to return to sporting activity after contracting the SARS-CoV-2 infection [9]. This protocol includes clinical-instrumental examinations to investigate the athletes’ cardiopulmonary functions, evaluating only their physical health and not their mental health, although the latter has a strong impact on the athletes’ performance.

A systematic review and meta-analysis performed in November 2022 highlighted that, during the COVID-19 pandemic, the prevalence of anxiety was 34.8% and the prevalence of depression was 32.4% [10]. Furthermore, many recent studies have showcased mental health consequences in patients who have previously suffered from COVID-19, as highlighted by the scoping review by Shanbehzadeh et al. [11]. Given the mental health consequences highlighted in recent scientific literature during the COVID-19 pandemic and in people who suffered from COVID-19, combined with the attention that mental wellbeing for professional athletes has been gaining all over the world in recent years, we decided to investigate how mental health has been affected in athletes who suffered from COVID-19 [12,13]. As mentioned above, in recent years, an increased importance has been placed on the mental health of professional athletes, in order to understand the conditions of high psychological, physical and social strain experienced by elite athletes, due to the high public interest in their performances and results [14]. In this context, the specific discipline of sports psychiatry and psychotherapy has focused on the diagnosis, treatment and prevention of psychological pathologies in athletes, aiming to safeguard their psychophysical wellbeing [15].

Studies aiming to investigate mental health outcomes in professional athletes after SARS-CoV-2 infection are performed through the use of different questionnaires, such as the *Athlete Psychological Strain Questionnaire* (APSQ) [16], or the *Depression Anxiety Stress Scales-21* (DASS-21) [17,18]; this last scale is not specific to athletes, but it has been shown that reliability is maintained when assessing psychological wellbeing outcomes in elite athletes [19].

The aim of this systematic review is to gather and summarize the current scientific evidence regarding the psychological sequelae and mental health of elite athletes who have been infected with the virus SARS-CoV-2.

## 2. Materials and Methods

A systematic review was performed following the Preferred Reporting Items for Systematic Reviews and Meta-Analyses (PRISMA) Statement [20]; three databases were searched: PubMed, ISI Web of Knowledge, and Scopus. Language (English language only) and date (from January 2020 to October 2022) filters were used for each database, as well as the title-abstract-keywords filter for Scopus.

The query used to perform the search was structured using PICO methodology, and contained keywords identifying professional athletes (P, population), having suffered from COVID-19 (I, intervention), and mental health outcomes (O, outcome); a control (C) was not included in the review. The following query was used for the search:
(Covid19 OR “COVID-19” OR “SARS-CoV-2” OR Covid OR Coronavirus OR “long COVID” OR “long-COVID” OR “post COVID” OR “chronic COVID syndrome” OR “long haul COVID” OR “long hauler COVID” OR “long-haul COVID” OR “persistent COVID-19” OR “post-acute COVID syndrome” OR “post-acute COVID-19 syndrome” OR “post-acute sequelae of SARS-CoV-2 infection”)AND(“mental fatigue” OR “mental health” OR “depress*” OR “anxiety” OR “psychological stress” OR “chronic fatigue syndrome*” OR “Chronic Fatigue Disorder*” OR “Lassitude” OR “Post-Traumatic Stress Disorder*” OR “Muscle Pain” OR “Insomnia Disorder*” OR “Insomnia” OR “Weakness” OR “Muscle Weakness”)AND(athlet* OR sport OR agonistic OR professionist)

After retrieving the articles from all the selected databases, duplicate removal and the initial screening by title and abstract was performed through the Rayyan website [21], which allowed articles to be screened by each researcher individually, in order to reduce selection bias following triple-blind methodology.

### 2.1. Inclusion Criteria

Articles investigating mental health in athletes who suffered from COVID-19 after the acute illness, written in English language, published from the start of the pandemic (January 2020) up until the search was performed (October 2022) were included in the review.

### 2.2. Exclusion Criteria

Manuscripts were excluded if they did not investigate mental health in athletes after they suffered from COVID-19.

### 2.3. Data Extraction and Synthesis

Following the PRISMA model [20], two researchers (V.F.C. and M.F.R.) performed the data extraction from the included articles, reporting results in an Excel worksheet. One review author (V.F.C.) extracted data from included studies and the second author (M.F.R.) checked the extracted data. Disagreements were resolved by a consensus-based discussion between the two review authors (V.F.C. and M.F.R.); if no agreement could be reached, it was planned a third author (C.A.) would intervene. The information to be extracted was established a priori and included: authors, year and country the study was conducted in, study design, follow-up time, athletes’ sport, sample size, number of athletes in whom depression, stress, or anxiety were diagnosed, and questionnaire used to assess mental health.

## 3. Results

According to the PRISMA flowchart (Figure 1), the initial search resulted in 2420 manuscripts from the three databases: 983 articles from PubMed, 960 articles from ISI Web of Knowledge and 477 articles from Scopus. After duplicate removal, 2379 articles were screened by title and abstract using the Rayyan website [21] to facilitate using the triple-blind methodology when performing the initial screening. Subsequently, the researchers screened 41 articles by full-text and 37 of these were excluded: 27 by wrong outcome, 8 by wrong population and 2 by wrong publication type. A total of 4 eligible articles were included in the systematic review after full-text screening. Any conflict about the inclusion or exclusion of the articles was resolved by internal discussion between the researchers.

Out of the 4 included studies, three were cross-sectional studies performed in Turkey [22,23,24], and one was a cohort study performed in Hungary [25]. Two of the included studies focused on football (English football, or soccer) players [22,23] and the other two included athletes from multiple disciplines [24,25]. The main characteristics of the included articles are summarized in Table 1.

All included articles measured depression in athletes who had suffered from COVID-19. Three articles reported 38.30% [22], 29.12% [23], and 64.88% [24] of athletes suffering from depression, whilst the fourth study [25] reported that, on average, 9.89% (SD ± 1.6) of athletes suffered from depression (the number was not showcased).

Three of the included studies measured stress in athletes, and highlighted 23.88% [22], 25.29% [23], and 70.83% [24] of athletes exhibiting stress after having suffered from COVID-19.

Concerning anxiety, all four studies investigated this outcome: three studies reported 30.84% [22], 34.48% [23], and 52.98% [24] of athletes suffering from anxiety, while the fourth [25] only reported the average, with 2.4% (SD ± 6.6) suffering from anxiety.

Overall, the only two questionnaires used in more than one study were the DASS-21 [17,18,19] and the APSQ [16] questionnaires, both used in two of the included studies.

## 4. Discussion

SARS-CoV-2 infection had a strong impact on the physical and mental health performance of elite athletes: in this systematic review, we investigated the impact of SARS-CoV-2 infection on psychological and mental health in this category of athletes.

In relation to the physical sequelae, the cardiological consequences appear to have a great impact on the athlete’s quality of life. Modica G. et al. [26], in their systematic review, showed that the prevalence (1% to 4%) of myocarditis in elite athletes should be considered in order to maintain adequate athlete performance.

The results of the studies among infected athletes included in our review showed that depression had a prevalence ranging from 9.89% (SD ± 1.6) to 64.88%, stress had a prevalence between 23.88% and 70.83% and anxiety ranged from 2.4% (SD ± 6.6) to 52.98%. According to these statements, the occurrence of such symptoms in athletes with a previous SARS-CoV-2 infection is worrying.

In relation to the psychological and mental sequelae, the depression appears to have a great impact on the athlete’s quality of life. As the results of our study show, all included articles measured depression and anxiety in athletes who had suffered from COVID-19, while stress was evaluated in three of included studies. Elite athletes are used to dealing with stress and anxiety, as situations that they encounter daily that are related to their training mode and competitive events [27]. Clemente-Suarez et al. [28] reported a higher level of anxiety in Olympic athletes during COVID-19, but in line with non-pathological conditions. This is due to the greater experience of high-performance athletes in dealing with competition-related anxiety and the existence of greater cognitive resources. Athletes’ repeated exposure to physical exercise may have led to a positive stress response system. Mehrsafar et al. [29] showed significant positive correlations between COVID-19 anxiety and somatic competitive anxiety, cognitive competitive anxiety and competition response.

A recent study by Lima et al., (2022) shows high scores of depression, anxiety, stress and psychological distress in athletes with previous infection with COVID-19, compared to athletes who have not contracted the virus [23]. While anxiety and stress may be symptoms related to an everyday condition of elite athletes and intensified during infection, depression is a symptom that needs particular attention. It is evident from our results that this condition has a high prevalence in the population analyzed; this may be due to both depression and anxiety being alleviated by involvement in their sporting excellence and athletic identity [30], lacking during the period of infection, combined with isolation which may lead to concerns about maintaining optimal physical performance with the thought of having to resume competitive activity after a period of total inactivity [31,32].

Of the four articles included in our review, Lima Y. et al. [23] found a gender difference in stress and depression levels, which increased in women compared to men. In agreement with our findings, Bucciarelli V. et al. confirm that women experience more severe and persistent mental health symptoms after COVID-19 infection and are at a higher risk of long COVID than men [33].

In addition, two of the studies included [23,24] have correlated sleep disorders with mental problems such as depression, anxiety and stress, in accordance with a prepandemic study by Biggins et al. [34]

The consequences of COVID-19, which have an impact on the athlete’s physical and mental health, also affect their working life. The systematic review by Gualano M.R. et al. [35] studied the impact of post-COVID-19 on previously hospitalized patients, showing that not all workers return to work and returning workers may be restricted in their working hours or activities.

This review has some strengths and limitations. A systematic approach was used to search the three chosen databases, following the PRISMA guidelines, and the selection of articles was carried out by three blind researchers. It is important to note that the use of different questionnaires to measure psychological wellbeing outcomes in elite athletes made it difficult to harmonize the results. Moreover, due to the use of different questionnaires, a meta-analysis was not performed.

## 5. Conclusions

The prevalence of psychological and mental health-related symptoms in individuals increased during the COVID-19 pandemic. Depression, anxiety and stress, when occurring in individuals from the general population, are generally attended to by specialists and frequently require specific treatment or pharmacological therapies. Consequently, we consider it necessary to have a specialist evaluation of whether these symptoms emerge in elite athletes as well. The results of our review, and the low number of studies currently present, show that mental health in athletes with a history of SARS-CoV-2 infection is an issue that requires more investigation, considering the numerous instances of evidence of clinical consequences. The importance of post-infection psychological sequelae is significant in order to assess possible repercussions on the athletes’ sporting performance and is recommended to promote prevention strategies in this population group.

## Figures and Tables

**Figure 1 ijerph-19-16377-f001:**
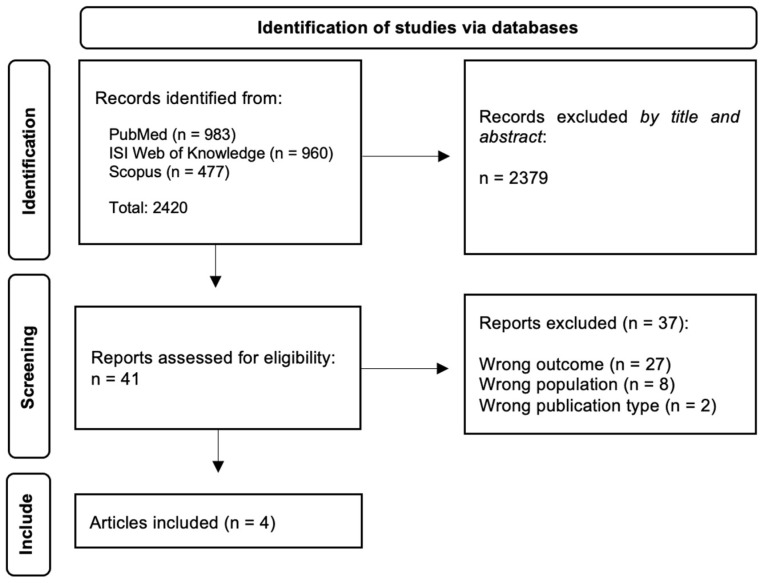
PRISMA flowchart.

**Table 1 ijerph-19-16377-t001:** Main characteristics of the included studies.

Authors	Year	Country	Study Design	Sport	Sample Size	Depression (N, or % and SD)	Stress (N)	Anxiety (N, or % and SD)	Mental Health Questionnaires Administered
Lima Y. et al. [22]	2021	Turkey	Cross-sectional	Football	201	77	48	62	DASS-21 ^1^, K-10 ^2^
Lima Y. et al. [23]	2022	Turkey	Cross-sectional	Football	261	76	66	90	APSQ ^3^, DASS-21 ^1^
Lima Y. et al. [24]	2022	Turkey	Cross-sectional	Multiple	168	109	119	89	APSQ ^3^, GAD-7 ^4^, PHQ-9 ^5^
Ocsovszky Z. et al. [25]	2022	Hungary	Cohort	Multiple	47	9.89% (SD) ± 1.6		2.4% (SD ± 6.6)	BDI ^6^, BAI ^7^, LS ^8^, MSPSS ^9^, PTSD test ^10^

^1^ Depression Anxiety Stress Scale, Short Version; ^2^ Kessler Psychological Distress Scale; ^3^ Athlete Psychological Strain Questionnaire; ^4^ General Anxiety Disorder-7; ^5^ Patient Health Questionnaire-9; ^6^ Beck Depression Inventory; ^7^ Beck Anxiety Inventory; ^8^ Life satisfaction; ^9^ Multidimensional Scale of Perceived Social Support; ^10^ Post traumatic stress disorder test (a shortened version of the original questionnaire was used).

## Data Availability

Not applicable.

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
