# Peer review of "Psychological and Mental Sequelae in Elite Athletes with Previous SARS-CoV-2 Infection: A Systematic Review"

_ijerph, 2022, doi:10.3390/ijerph192416377_

Round 1
Reviewer 1 Report
The article presented meets the requirements to be admitted, however a series of suggestions are made so that the authors include these aspects in their work or eliminate some other aspects.
It does not make sense that one of the selection criteria of the selection and search is the works written in English and Italian. Why if in Italian and not in other languages. The authors are from that context, but it is not justified that it be in Italian and why not in French, Spanish or another language. The display of international works takes the English language as a reference. Therefore, they should eliminate the search criteria for jobs in Italian, and keep only the English language.
In addition, none of the works selected in this systematic review (n=4) is an Italian work, there are 3 Turkish studies and one Hungarian, the Turkish studies being from the same group of authors (Lima et al.).
It would be necessary to reflect on whether a systematic review with such restrictive criteria is so precise, to obtain only 4 records, two of them being related to football and the other 2 contemplating different disciplines, which can be very diverse, it is not the same to compare results of individual sports, with team sports, racket sports, etc.
For the rest, the article in a few pages presents what is necessary in this type of article. As only 4 records were analyzed, little evidence was obtained and therefore the ability to generalize the results is low to other contexts, as the search is so restricted, although it is undoubtedly specialized.
Author Response
Dear Reviewer, thank you for your comments, which helped us to make the article better. We have provided a point-by-point response to your concerns regarding the manuscript.
Please see the attachment.

Author Response
Dear Reviewer,
Thank you for agreeing to review our manuscript. We have tried to improve our manuscript according to the revisions provided.
Please see the attachment.

Reviewer 3 Report
The theme this paper addresses is essential for science, especially in the context of the Covid-19 pandemic, which distressed all mankind in an almost brutal way. Sports were not an exception and, as it happened in so many other activity fields, there were extensive consequences in this area. Having said that, I will make my comments regarding this article, hoping that this input will be taken into account as positive criticism.
The abstract describes in detail the selection process of the articles which have become part of the systematic review, but it's less clear about the consequences of the analysis the authors performed. What is the added value of this research for the academic community?
The introduction is mostly addressing medical issues, causes, and consequences of Covid infection, but only one paragraph regarding mental health issues. This section should be particularly extended since the results of the analysis are evoking the psychological aspects and consequences.
The Method of selection is extensively described, and inclusion and exclusion criteria are enumerated. But I feel the authors fail to clearly explain why these specific criteria were used. Since out of 2420 articles only 4 were eligible, I have to wonder, besides the fact that the literature is scarce in this field, what other important aspect does this paper conclude? Table 1 (page 4) shows that in fact, only two authors studied the psychological sequelae and mental health of elite athletes who have been infected by Covid-19. Furthermore, besides football, which other sports have been investigated? How is this information relevant to the study? The authors state at some point some percentages, like 75% or 25% of studies. But, in fact, 75% means 3 studies. It is not the most appropriate manner to present cohorts that contain only one, two, or three elements.
The authors discuss depression and anxiety as consequences of the infection with Covid-19 in athletes but fail to argue what is the importance of their discoveries.
The conclusion is not substantial and fails to evidentiate and support the relevance of this study. Perhaps the field of study and the population has certain particularities, it is not clear from the discussion or conclusions.
Author Response
Dear Reviewer,
thank you for your opinions and comments which enabled us to make the article more comprehensible, with respect to the purpose for which it was written. We have provided a point-by-point response to your concerns regarding the manuscript.
Please see the attachment.

Round 2
Reviewer 3 Report
The authors took into consideration the suggestions previously made by the reviewers, addressing the issues reviewers suggested. It is still not clear in section 2.3 Data extraction and synthesis, what discrepancies were taken into consideration when selecting the articles that were analyzed.
Author Response
We thank the reviewer for the clarification. We have tried to further implement section 2.3 "Data extraction" (lines 130-139) by modifying it as follows:
“Following the PRISMA model [20], two researchers (V.F.C. and M.F.R.) performed the data extraction from the included articles reporting results in an Excel worksheet. One review author (V.F.C.) extracted data from included studies and the second author (M.F.R.) checked the extracted data. Disagreements were resolved by a consensus-based discussion between the two review authors (V.F.C. and M.F.R.); if no agreement could be reached, it was planned a third author (C.A.) would intervene. The information to be extracted was established a priori and included: authors, year and country the study was conducted in, study design, follow-up time, athletes’ sport, sample size, number of athletes in whom depression, stress, or anxiety were diagnosed, and questionnaire used to assess mental health.”
The main reasons for disagreement were about the inclusion of specific symptoms, only if related to previous SARS-CoV-2 infection, or study design.